# Is this the Right Neighborhood? Accurate and Query Efficient Model Agnostic Explanations

**Amit Dhurandhar**
IBM Research
Yorktown Heights, NY, USA
adhuran@us.ibm.com

**Karthikeyan Natesan Ramamurthy**
IBM Research
Yorktown Heights, NY, USA
knatesa@us.ibm.com

**Karthikeyan Shanmugam**\*
IBM Research
Yorktown Heights, NY, USA
karthikeyanshanmugam88@gmail.com

## Abstract

There have been multiple works that try to ascertain explanations for decisions of black box models on particular inputs by perturbing the input or by sampling around it, creating a neighborhood and then fitting a sparse (linear) model (e.g. LIME). Many of these methods are unstable and so more recent work tries to find stable or robust alternatives. However, stable solutions may not accurately represent the behavior of the model around the input. Thus, the question we ask in this paper is are we approximating the local boundary around the input accurately? In particular, are we sampling the right neighborhood so that a linear approximation of the black box is faithful to its true behavior around that input given that the black box can be highly non-linear (viz. deep relu network with many linear pieces). It is difficult to know the correct neighborhood width (or radius) as too small a width can lead to a bad condition number of the inverse covariance matrix of function fitting procedures resulting in unstable predictions, while too large a width may lead to accounting for multiple linear pieces and consequently a poor local approximation. In this paper, we propose a simple approach that is robust across neighborhood widths in recovering faithful local explanations. In addition to a naive implementation of our approach which can still be accurate, we propose a novel adaptive neighborhood sampling scheme (ANS) that we formally show can be much more sample and query efficient. We then empirically evaluate our approach on real data where our explanations are significantly more sample and query efficient than the competitors, while also being faithful and stable across different widths.

## 1 Introduction

Explainable artificial intelligence (XAI) has come to prominence in recent years with the proliferation of deep learning technologies, which are inherently black box, across various facets of society [13]. Regulations such as the General Data Protection Regulation (GDPR) [31] in Europe demand explanations to be provided for automatic decision making systems that affect humans. The explanations demanded are typically local in the sense that we want explanations for individual decisions rather than for the entire system, which could also be much more challenging. Given this need many local

---

\*Author's current affiliation is Google Research India. Work was done while at IBM Research.

36th Conference on Neural Information Processing Systems (NeurIPS 2022).

explanation methods have been developed [26, 21, 6, 24, 33] that can be used to explain arbitrary models, i.e. are model agnostic. Even though a plethora of these methods exist it is not clear if they are truly faithful to the underlying black box model. Most of these methods employ some type of sampling or perturbation scheme to estimate a simple interpretable model (viz. a sparse linear model), which can then be read off to ascertain explanations. Because of the inherent randomness of such procedures the most important question is are we learning the right (local) interpretable model? This essentially boils down to *are we using the right samples for the estimation?*

In this paper, we argue that sampling a neighborhood just based on the input values and using it for estimation can lead to a poor local approximation of the black box function. This also applies when samples are generated from an underlying data manifold [2], which is oblivious to the black box's behavior. As such, we propose a novel neighborhood sampling scheme called *Adaptive Neighborhood Sampling* (ANS) which generates a neighborhood taking into account the local behavior of the black box. Our main idea is to estimate the region where the black box model is (approximately) linear around the input we want to explain followed by adaptively sampling in this region still respecting the original sampling process/distribution. The estimate of the relevant linear region will be done using much fewer black box queries than say a total budget of $N$. As it will turn out our sampling process will result in many more samples in the desired region hence minimizing *query wastage* compared with simply querying the black box $N$ times, finding the relevant region and then fitting a sparse linear model. Query efficiency is important as it has been argued in recent works [5, 18] that in todays multi-cloud environments each query can have an associated monetary cost not to mention factors such as inference time, power consumption and network latency in a distributed system need to be taken into account.

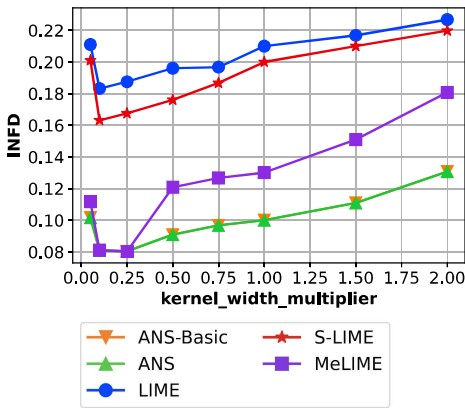

Figure 1: We show the variation of Infidelity (INFD) measure over various kernel width multiplier values for the proposed methods (ANS and ANS-Basic) with CIFAR10 dataset. Lower values of this measure are better. At very small kernel widths, INFD actually goes up indicating instability. Hence in lieu of choosing arbitrarily low kernel widths, we need explanations that are accurate over many kernel widths, which is provided by our methods. More details are in the experiments section.

An illustration of our approach is shown in Figure 1. While it can be argued that very small kernel widths, which corresponds to considering only extremely close samples in the sparse linear estimation, can be chosen to circumvent the problem of choosing the right neighborhood, we show that this can lead to instability (high infidelity). Hence it is important to find methods that work well across multiple kernel widths. This is achieved by our methods (ANS, ANS-Basic). In addition, choosing the right neighborhood also leads to high sample and query efficiency for ANS.

In the rest of the paper, we repeatedly use the terms faithful and stable with respect to the explanations. What we mean by faithful is that simple models (viz. sparse linear) built using these explanations should closely replicate the behavior of the black box model on the corresponding examples. This corresponds to low infidelity (INFD). By stability (unless mentioned otherwise) we mean for different kernel widths we should recover similar quality explanations. This corresponds to lesser variation in INFD and coefficient inconsistency (CI), both of which are defined in Section 4, across kernel widths.

## 2   Related Work

There have been many works trying to uncover local explanations in model agnostic settings [26, 33, 6, 21, 25]. Local explanations also do not have to be feature based and could also be exemplar based [14]. There are also methods that provide both forms of explanations [24]. Some methods in a sense stitch together local explanations and create global ones [25, 23]. Although there are many methods there is still a need to design explanation methods that are faithful and stable [12]. This is

particularly true for LIME like neighborhood sampling methods where stability and faithfulness w.r.t. neighborhood width is highly desired [32] as it is difficult to predefine an optimal width.

Some recent works try to accomplish improved stability for certain modalities such as tabular [15] through adversarial training. This notion of stability however is across examples sampled from linear shifts in the distribution. Others assume access to a partial causal graph [10, 17] and build on top of SHAP to create stable feature attributions. Stability here is achieved by trying to obtain attributions that are as consistent as possible with respect to the provided partial graph. Another popular strategy is to learn the data manifold and then sample from it [2, 29] or project onto it [1] creating more faithful explanations. The premise here is that the realistic sampled neighbors will give rise to better explanations. There are other works that aim for stability and unidirectionality [6] in the explanations, by proposing a new algorithm for fitting rather than modifying the sampling scheme. Hence the methodology in [6] is complimentary in a sense to our contribution and could potentially benefit from the neighborhood sampling scheme proposed in our current work.

The conceptual contribution of our work is to realize that manifold or not, examples sampled in a neighborhood could belong to different linear parts in a non-linear function such as a deep relu network and hence using them to obtain a local explanation through sparse linear or some other simple model fitting can be misleading.

---

**Algorithm 1:** Adaptive neighborhood sampling (ANS). Estimation of $a$ and $b$ can also be performed not just once but multiple times and the latest estimates can be used for future sampling. For realistic perturbations sampling can be done in the latent space. More details in section 3.1.

**Input:** Example to explain $\mu$, black-box predictor $f(.)$, maximum number of neighbors generated $N$, standard deviation $\sigma$ and number top features to output $k$

Set $Q = \phi$    # Examples to query
Sample $n$ $(<< N)$ examples from $\mathcal{N}(\mu, \sigma^2 I)$ and query $f(.)$
Find region $[a_n, b_n]$ that corresponds to $\mu$ using MPLSR methods [9, 7] on the $n$ samples
Add to $Q$ samples that lie in $[a_n, b_n]$
Estimate uncertainty $\alpha$    # Could be set $\propto \frac{1}{\sqrt{n}}$ or based on stability of the region (i.e. $1 - \rho$)
Sample $N - n$ examples from $\mathcal{N}\left(\alpha\mu + (1 - \alpha)\frac{a_n + b_n}{2}, \sigma^2 I\right)$
Add to $Q$ samples that lie in $[a_n, b_n]$
Query $f(.)$ on these additional samples added to $Q$
Fit interpretable model (viz. sparse linear) $l(.)$ to $(x, f(x))$ where $x \in Q$

**Output:** Top $k$ coefficients of $l(.)$

---

# 3 Method

We now describe our approach. In terms of notation vectors are bolded. Matrices are also bolded and in capital letters. Inequalities and assignments indicated for vectors and matrices are element wise.

## 3.1 Description

As mentioned before, if you consider the black box function to be piecewise linear (viz. deep relu network) our main idea is to sample a neighborhood around an input $\mu$ such that the neighbors belong to the linear piece corresponding to it. Since any continuous black box function can be arbitrarily well-approximated by a piecewise linear function [20] and hence our setup is quite general.

Typical neighborhood sampling schemes such as those for LIME and its variants [2, 33, 6] (i.e. ambient or latent space) can be seen to sample from an independent Gaussian distribution $\mathcal{N}(\mu, \sigma^2 I)$, where $\sigma$ is the standard deviation for each feature (or kernel width) and $\mu$ is a $d$ dimensional example we want to explain. It is difficult to find the correct $\sigma$ as too small a $\sigma$ can lead to numeric instabilities during function fitting because of a bad condition number of the inverse covariance matrix, while too large a $\sigma$ will not accurately capture the local information [32]. So a pertinent question is, *can we*

*devise a neighborhood sampling procedure that captures local information and is (largely) robust across different values of $\sigma$?*

**Main Idea:** Our idea is to leverage multidimensional piecewise linear segmented regression (MPLSR) schemes [9, 7] to assist in this endeavor. These methods will learn a piecewise linear function identifying boundaries of each linear piece. One simple (or naive) solution for our problem using these methods would be to i) sample $N$ neighbors using $\mathcal{N}(\boldsymbol{\mu}, \sigma^2 \boldsymbol{I})$, ii) run MPLSR on these samples to find the boundaries of the linear piece corresponding to $\boldsymbol{\mu}$, iii) select only samples that belong to this region and iv) train a sparse linear or some other interpretable model only on the selected samples followed by outputting the important features. Although this approach captures our general idea, it has two sources of inefficiency; i) *Low sample efficiency:* If $\boldsymbol{\mu}$ is close to the boundary of the range $[\boldsymbol{a}, \boldsymbol{b}]$ in which it lies, then the naive sampling scheme will have many samples lying outside the range leaving much fewer valid samples to train an interpretable model. ii) *Low query efficiency:* The naive scheme requires us to query the black box $N$ times, but only a small fraction of these samples may lie within the desired range, as previously mentioned.

**Algorithm 1:** Given this, in algorithm 1 we propose an adaptive neighborhood sampling (ANS) scheme which aims to enhance both these efficiencies. A detailed analysis is given in Section 3.2. Our idea is to sample a small set of examples $n$ similar to the naive scheme. However, we then run MPLSR and find an estimate of the true range $[\boldsymbol{a}, \boldsymbol{b}]$ given by $[\boldsymbol{a_n}, \boldsymbol{b_n}]$. Depending on our uncertainty in the estimate denoted by $\alpha \in [0, 1]$ we sample from an updated distribution $\mathcal{N}\left(\alpha\boldsymbol{\mu} + (1 - \alpha)\frac{\boldsymbol{a_n} + \boldsymbol{b_n}}{2}, \sigma^2 \boldsymbol{I}\right)$. The intuition behind the updated distribution is that if we are highly uncertain (i.e. $\alpha \approx 1$) about the range then we want to be conservative and sample close to $\boldsymbol{\mu}$. However, if we are confident $\alpha \approx 0$, then we want to sample close to the middle of the range so that we get (many) valid samples belonging to the correct linear piece that corresponds to $\boldsymbol{\mu}$.

For realistic perturbations using autoencoders one can sample in the latent space, find the (linear) region in the input space by decoding the current neighboring samples followed by updating the mean of the sampling distribution as indicated in algorithm 1 and then passing it through the autoencoder to find its corresponding latent to sample around.

**Estimating $\alpha$:** $\alpha$ could be estimated in at least a couple of ways: i) Since the error of MPLSR [7] schemes in terms of $n$ scales as $\frac{1}{\sqrt{n}}$, we could set $\alpha \propto \frac{1}{\sqrt{n}}$. ii) One could also perform subsampling or bootstrap sampling on the $n$ samples[2], run MPLSR on each such sample and compute a (normalized) metric indicative of the variability in estimating the range $[\boldsymbol{a}, \boldsymbol{b}]$. For instance, one could use the overlap coefficient [30] $\rho$ as an indicator of the stability of the range. Given $m$ bootstrap samples with corresponding estimated ranges using MPSLR being $[\boldsymbol{a_n^{(1)}}, \boldsymbol{b_n^{(1)}}], ..., [\boldsymbol{a_n^{(m)}}, \boldsymbol{b_n^{(m)}}]$ and their corresponding volumes being $\vartheta_1, ..., \vartheta_m$ respectively, if $\vartheta_\cap$ denotes the volume of the intersection of these ranges (i.e. volume of $[\boldsymbol{a_n^{(1)}}, \boldsymbol{b_n^{(1)}}] \cap ... \cap [\boldsymbol{a_n^{(m)}}, \boldsymbol{b_n^{(m)}}]$), then the overlap coefficient would be

$$\rho = \frac{\vartheta_\cap}{\min\{\vartheta_1, ..., \vartheta_m\}} \tag{1}$$

The coefficient would be indicative of how stably we can estimate the range, where if we get the exact same ranges for the $m$ bootstraps, $\rho$ will be 1, while if the ranges had little overlap, $\rho$ would be close to 0. Our uncertainty $\alpha$ can then be set to $1 - \rho$.

## 3.2 Analysis

We now analyze our approach in terms of sampling efficiency, query efficiency and time complexity.

### 3.2.1 Sampling Inefficiency

The sampling inefficiency or the expected number of iterations to get a valid sample in the range $[\boldsymbol{a}, \boldsymbol{b}]$ based on a naive implementation of our idea which is described in the previous subsection (lets call it ANS-Basic) that involves sampling from the original distribution $\mathcal{N}(\boldsymbol{\mu}, \sigma^2 \boldsymbol{I})$ would be $\frac{1}{P(\boldsymbol{x} \in [\boldsymbol{a}, \boldsymbol{b}])}$. If we sample $n$ examples based on the original distribution then the sample efficiency will be the

---

[2]Note this will not require additional queries to the black box.

same for these examples. However, for $N - n$ examples if we sample using the proposed distribution $\mathcal{N}\left(\alpha\boldsymbol{\mu} + (1 - \alpha)\frac{\boldsymbol{a_n} + \boldsymbol{b_n}}{2}, \sigma^2\boldsymbol{I}\right)$ the sampling inefficiency will be $\frac{1}{P_\alpha(\boldsymbol{x} \in [\boldsymbol{a_n}, \boldsymbol{b_n}])} \cdot \frac{P_\alpha(\boldsymbol{x} \in [\boldsymbol{a_n}, \boldsymbol{b_n}])}{P_\alpha(\boldsymbol{x} \in [\boldsymbol{a}, \boldsymbol{b}])} = \frac{1}{P_\alpha(\boldsymbol{x} \in [\boldsymbol{a}, \boldsymbol{b}])}$, where $P_\alpha(.)$ denotes the (cumulative) probability w.r.t. the proposed distribution. Thus, the relative sampling inefficiency $s_I$ will be as follows:

$$
\begin{aligned}
s_I &= \frac{\text{Inefficiency of ANS}}{\text{Inefficiency of ANS-Basic}} \\
&= \frac{\frac{n}{N}\frac{1}{P(\boldsymbol{x} \in [\boldsymbol{a}, \boldsymbol{b}])} + \frac{N-n}{N}\frac{1}{P_\alpha(\boldsymbol{x} \in [\boldsymbol{a}, \boldsymbol{b}])}}{\frac{1}{P(\boldsymbol{x} \in [\boldsymbol{a}, \boldsymbol{b}])}} \\
&= \frac{n}{N} + \frac{N - n}{N}\frac{P(\boldsymbol{x} \in [\boldsymbol{a}, \boldsymbol{b}])}{P_\alpha(\boldsymbol{x} \in [\boldsymbol{a}, \boldsymbol{b}])}
\end{aligned}
\tag{2}
$$

When $\alpha = 1$ (i.e. high uncertainty of range) $P_\alpha(\boldsymbol{x} \in [\boldsymbol{a}, \boldsymbol{b}]) = P(\boldsymbol{x} \in [\boldsymbol{a}, \boldsymbol{b}])$ and hence our efficiency is the same as sampling from the original distribution. However, for smaller $\alpha$, $P_\alpha(.)$ should significantly increase making our procedure much more sample efficient especially when the range is large and $\boldsymbol{\mu}$ is far away from the center of the range. Hence,

$$
\ln\left(\frac{P(\boldsymbol{x} \in [\boldsymbol{a}, \boldsymbol{b}])}{P_\alpha(\boldsymbol{x} \in [\boldsymbol{a}, \boldsymbol{b}])}\right) \propto -\frac{(1 - \alpha)}{\sigma^2}||\boldsymbol{b} - \boldsymbol{a}||\left|\left|\frac{\boldsymbol{a} + \boldsymbol{b}}{2} - \boldsymbol{\mu}\right|\right|
\tag{3}
$$

Ideally, we would want to choose an $n$ where equation 2 is minimized. Finding the optimal $n$ in closed form is difficult. Note that there is a tradeoff for different values of $n$. Too large a $n$ will lead us to sample very few points from our proposed distribution thus mitigating its benefit. Too small a $n$ might lead to high uncertainty (i.e. $\alpha$ close to 1) in estimation of the range $[\boldsymbol{a}, \boldsymbol{b}]$ which will again increase $s_I$. However one could in practice choose a reasonable $n$ depending on the dimension of the data. One could also evaluate the quality by looking at (random) subsets or bootstrapped samples of the queried $n$ instances, fitting the linear piecewise methods and evaluating the stability of the obtained partitions, which is similar to one of the methods proposed to estimate $\alpha$.

Further, one does not only have to perform the estimation of the ranges once but can do so periodically updating the estimates for $\boldsymbol{a}$ and $\boldsymbol{b}$. The relative inefficiency analysis in this case also should naturally extend. For instance, if we estimate the ranges $r$ times producing the corresponding $\alpha_i$ during the $i^{\text{th}}$ estimation at intervals of $n_i$ respectively with $n_{r+1} = N - \sum_{i=1}^{r} n_i$ and $P_{\alpha_1}(\boldsymbol{x} \in [\boldsymbol{a}, \boldsymbol{b}]) = P(\boldsymbol{x} \in [\boldsymbol{a}, \boldsymbol{b}])$, then the relative inefficiency $s_I^{(r)}$ would be,

$$
s_I^{(r)} = \frac{1}{N}\sum_{i=1}^{r+1} n_i \frac{P(\boldsymbol{x} \in [\boldsymbol{a}, \boldsymbol{b}])}{P_{\alpha_i}(\boldsymbol{x} \in [\boldsymbol{a}, \boldsymbol{b}])}
\tag{4}
$$

where, $s_I^{(1)}$ can be denoted as simply $s_I$. We expect that we would get better estimates of the range for later iterations i.e. $\alpha_j \leq \alpha_i \ \forall j > i$ and hence $P_{\alpha_j}(\boldsymbol{x} \in [\boldsymbol{a}, \boldsymbol{b}]) \geq P_{\alpha_i}(\boldsymbol{x} \in [\boldsymbol{a}, \boldsymbol{b}])$ leading to higher efficiency for examples sampled at later stages. Of course the larger the $r$, more the computational complexity, as we have to run the piecewise linear routine $r$ times.

### 3.2.2 Query Efficiency

What we saw up until now was the efficiency in sampling examples within the range. Interestingly however, the (relative) black box *query* efficiency is likely to be much higher. The reason for this is that with ANS we need only query those examples that are accepted by our sampling scheme as opposed to querying all the examples as in the naive sampling case. The efficiency of naive sampling is thus still $P(\boldsymbol{x} \in [\boldsymbol{a}, \boldsymbol{b}])$, while for us the accepted examples after sampling $N - n$ times which in expectation will be $(N - n)P_\alpha(\boldsymbol{x} \in [\boldsymbol{a_n}, \boldsymbol{b_n}])$ will definitely lie within $[\boldsymbol{a_n}, \boldsymbol{b_n}]$ and hence the probability they will lie in $[\boldsymbol{a}, \boldsymbol{b}]$ is given by $\frac{P_\alpha(\boldsymbol{x} \in [\boldsymbol{a}, \boldsymbol{b}])}{P_\alpha(\boldsymbol{x} \in [\boldsymbol{a_n}, \boldsymbol{b_n}])}$. Note that the region $[\boldsymbol{a}, \boldsymbol{b}]$ will not be underestimated based on finite samples i.e. $[\boldsymbol{a}, \boldsymbol{b}] \subseteq [\boldsymbol{a_n}, \boldsymbol{b_n}]$ given a good piecewise linear method, and hence the ratio of probabilities meaningfully represents the probability of accepted samples lying

in $[\boldsymbol{a}, \boldsymbol{b}]$. The reason for this is that our sampling can never add not existent non-linearities as this is just a function of the black box model, but may miss regions where the linear piece corresponding to $\boldsymbol{\mu}$ ends. Hence, the total relative query efficiency $q_E$ of ANS is,

$$
\begin{aligned}
q_E &= \frac{\frac{n}{N} P(\boldsymbol{x} \in [\boldsymbol{a}, \boldsymbol{b}]) + \frac{N-n}{N} \frac{P_\alpha(\boldsymbol{x} \in [\boldsymbol{a}, \boldsymbol{b}])}{P_\alpha(\boldsymbol{x} \in [\boldsymbol{a_n}, \boldsymbol{b_n}])}}{P(\boldsymbol{x} \in [\boldsymbol{a}, \boldsymbol{b}])} \\
&= \frac{n}{N} + \frac{N-n}{N} \frac{P_\alpha(\boldsymbol{x} \in [\boldsymbol{a}, \boldsymbol{b}])}{P_\alpha(\boldsymbol{x} \in [\boldsymbol{a_n}, \boldsymbol{b_n}]) P(\boldsymbol{x} \in [\boldsymbol{a}, \boldsymbol{b}])}
\end{aligned}
\tag{5}
$$

The gain can thus be large when we are quite certain of the range (i.e. small $\alpha$) for not too large a $n$ and the black box model is highly non-linear since, $\frac{P_\alpha(\boldsymbol{x} \in [\boldsymbol{a}, \boldsymbol{b}])}{P_\alpha(\boldsymbol{x} \in [\boldsymbol{a_n}, \boldsymbol{b_n}])}$ will be close to 1 and $P(\boldsymbol{x} \in [\boldsymbol{a}, \boldsymbol{b}])$ will in all likelihood be small. Similar to equation 4, here too the above analysis can be straightforwardly extended to multiple estimations of the range where the $\alpha$ would be replaced by the corresponding $\alpha_i$ for the $i^{\text{th}}$ estimate.

### 3.2.3 Time Complexity

The additional time complexity for ANS over LIME comes from having to run MPLSR methods [9, 7] which have complexities such as $O(nd \log n)$ or $O(nd^2)$. However, this time could be reduced across examples through embarrassing parallelism [16]. Moreover, efficient heuristics maybe used to speed up such partitioning schemes [11]. Interestingly though, with deeper models where inference time is not insignificant the lower query complexity of ANS can more than compensate for the additional time it requires running the MPLSR schemes. We see evidence of this in the experiments.

If we use $\rho$ to estimate $\alpha$ we may not have to estimate it per example, but we could estimate it for a few examples and then the maximum of those values (if we want to be conservative) could be used for other examples we want to explain for a given $n$. This will save on time as the amortized complexity over many examples will be similar to running MPSLR schemes just once per example to find the range $[\boldsymbol{a_n}, \boldsymbol{b_n}]$, rather than multiple times.

## 4 Experiments

We now empirically evaluate our approach. We compare ANS with ANS-Basic, LIME and Smoothed-Lime (S-Lime) on two commonly used tabular datasets IRIS [4] and HELOC [8], where the latter was the dataset used in the FICO explainability challenge. We also experiment on CIFAR10 [19] where an additional baseline Melime [2] is compared with. The black box models for the tabular datasets are random forest classifiers consisting of 100 trees that have an accuracy of $\sim 100\%$ and $\sim 78\%$ on IRIS and HELOC respectively, which is state-of-the-art. The black box model for CIFAR10 is a ResNet-18 which has an accuracy of $\sim 95\%$. Except CIFAR10 which comes with its own test we randomly split the other two datasets into 80% train and 20% test. For 10 randomly selected examples in each test set we performed bootstrap sampling 20 times for the respective $n$, estimated $[\boldsymbol{a_n}, \boldsymbol{b_n}]$ and calculated $\alpha$ based on the overlap coefficient $\rho$ as defined in equation 1. The maximum of those values was the $\alpha$ used for all the examples.

Given our goal of showcasing that the ANS (and ANS-Basic) neighborhood generation procedures elevate a given proxy model explanation scheme with generated neighborhoods we believe LIME, S-LIME and Melime to be good baselines which validate our idea on random as well as realistic neighborhoods. Methods such as SHAP [21] and MAPLE [24] do not really fit in here as they do not perform proxy model fitting on generated neighborhoods. Moreover, saliency based methods such as gradcam, saliency maps, layerwise relevance propagation also do not fit in our framework since they are white-box explanation methods.

**Metrics:** We now define evaluation metrics that we use to evaluate methods in this paper. The five metrics we use are Infidelity (INFD) [26], Generalized Infidelity (GI) [25], Coefficient Inconsistency (CI) [15], accepted sampling complexity (ASC) and query complexity (QC). INFD and GI evaluate fidelity of the explanations. CI evaluates stability. ASC and QC evaluate the sample and query efficiency respectively.

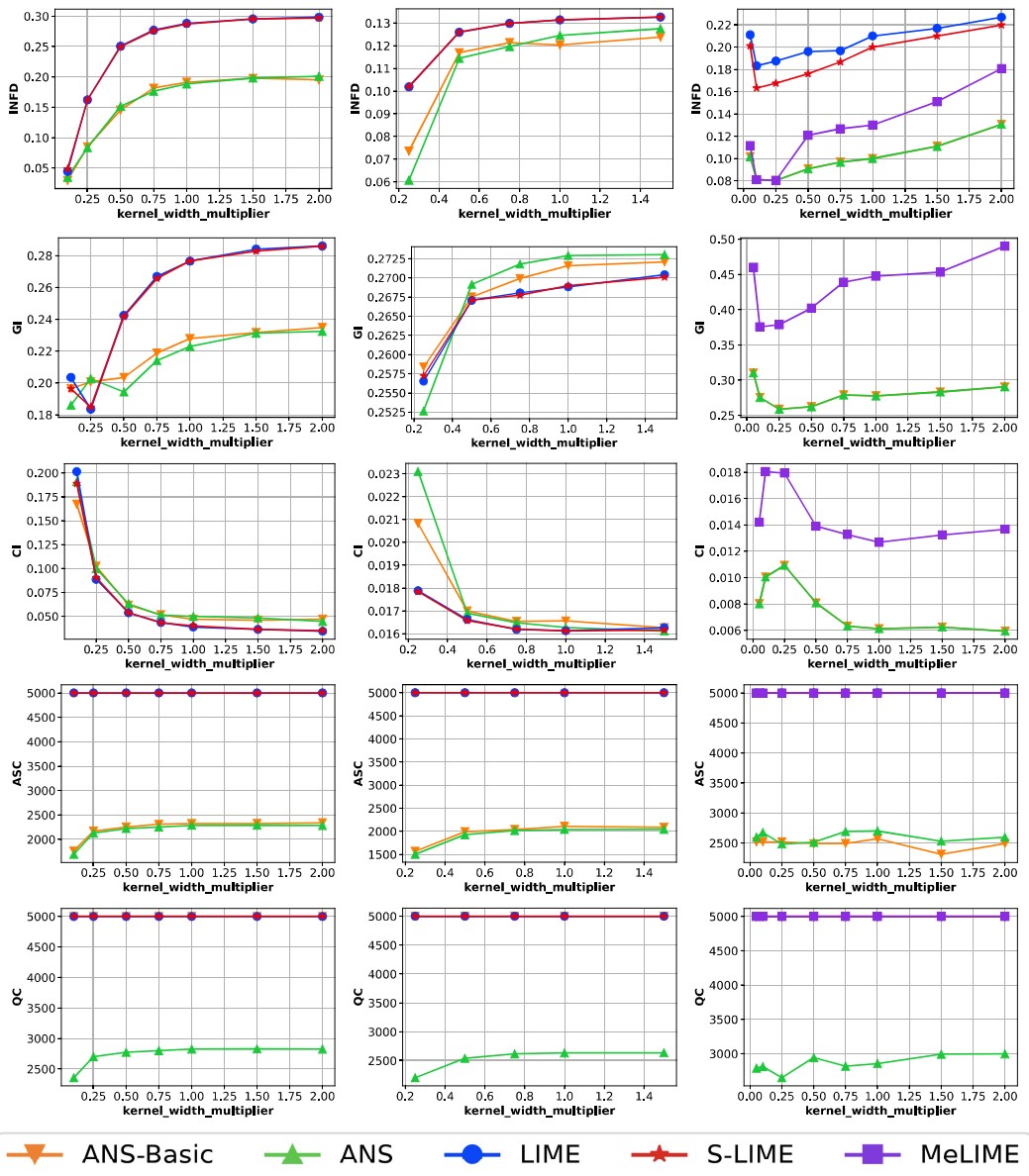

Figure 2: Various metrics vs. kernel width multiplier (kwm) for Iris (left), HELOC (middle), and CIFAR (right) datasets. The metrics are (from top to bottom - each row): Infidelity (INFD), Generalized Infidelity (GI), Coefficient Inconsistency (CI), Accepted Sample Complexity (ASC), and Query Complexity (QC). For all metrics lower values are better. The method legend is at the bottom.

If $(\boldsymbol{x}, \boldsymbol{y})$ denote examples in a test set $D_t$. Let $y_b(\boldsymbol{x})$ be the black box models prediction on an input $\boldsymbol{x}$ and $y_e^{\boldsymbol{x}'}(\boldsymbol{x})$ be the prediction on $\boldsymbol{x}$ applying the explanation model at $\boldsymbol{x}'$. Also let $c_e^{\boldsymbol{x}}$ and $\mathcal{N}_{\boldsymbol{x}}$ denote the feature attributions and the test/real neighborhood of $\boldsymbol{x}$ with $|.|_{\text{card}}$ denoting cardinality. With we have the following definitions.

*Infidelity (INFD):* We define infidelity as the mean absolute error (MAE) between the explanation and black box model over the test set. $\text{INFD} = \frac{1}{|D_t|_{\text{card}}} \sum_{(\boldsymbol{x}, \boldsymbol{y}) \in D_t} |y_b(\boldsymbol{x}) - y_e^{\boldsymbol{x}}(\boldsymbol{x})|$.

*Generalized Infidelity (GI):* Based on previous works we define GI as: $\text{GI} = \frac{1}{|D_t|_{\text{card}}} \sum_{(\boldsymbol{x}, \boldsymbol{y}) \in D_t} \frac{1}{|\mathcal{N}_{\boldsymbol{x}}|_{\text{card}}} \sum_{\boldsymbol{x}' \in \mathcal{N}_{\boldsymbol{x}}} |y_b(\boldsymbol{x}) - y_e^{\boldsymbol{x}'}(\boldsymbol{x})|$.

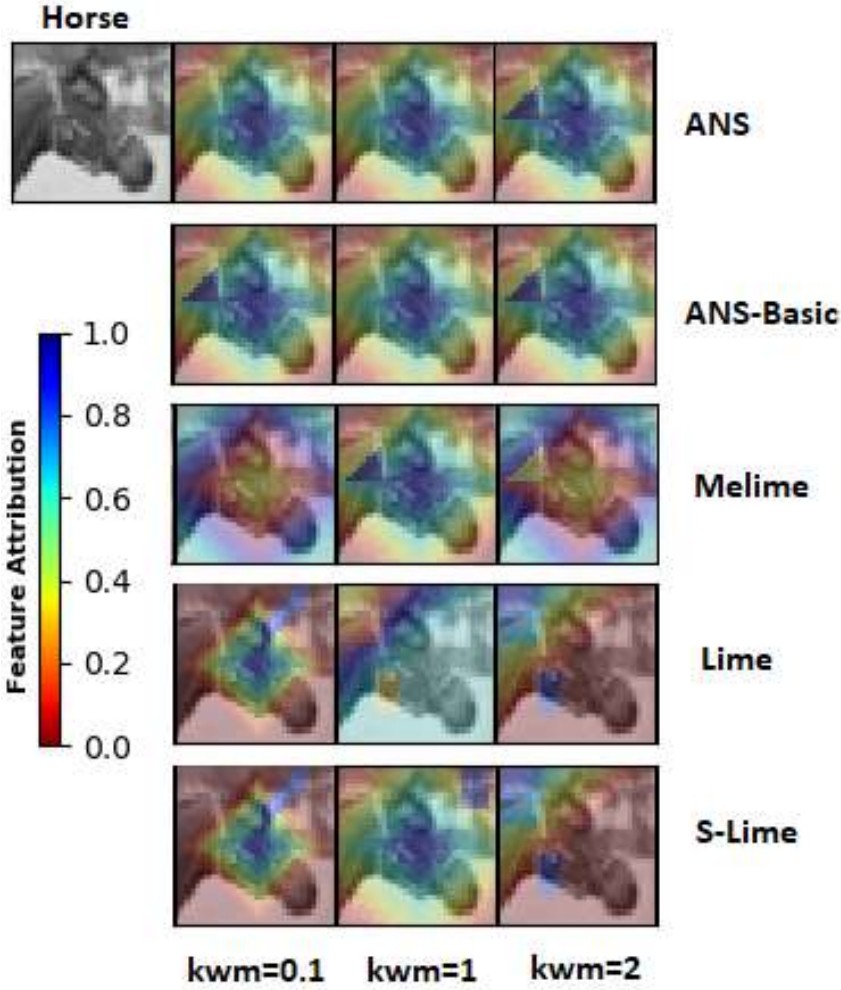

Figure 3: Qualitative explanations for an image from the CIFAR dataset for three different kernel width multiplier (kwm) values. The feature attributions of our proposed method ANS (and even ANS-Basic) are much more stable and accurate over different kwm values compared to other methods as the face of the horse is highlighted for different kwms. This is not the case for the other methods. More examples in the supplement.

*Coefficient Inconsistency (CI):* CI can be defined as the MAE between the attributions of the test points and their neighbors: $CI = \frac{1}{|D_t|_{\text{card}}} \sum_{(\boldsymbol{x},\boldsymbol{y}) \in D_t} \frac{1}{|\mathcal{N}_{\boldsymbol{x}}|_{\text{card}}} \sum_{\boldsymbol{x}' \in \mathcal{N}_{\boldsymbol{x}}} |c_e^{\boldsymbol{x}} - c_e^{\boldsymbol{x}'}|_1$.

*ASC:* This is just the total number of samples used to train the local sparse linear models. For ANS and ANS-Basic they are the accepted samples. While for other baselines it is just $N$.

*QC:* This is the total number of queries made by the methods to the black box model.

We report the results in Figure 2. The results show the behavior of the different metrics for different kernel width multiplier (kwm) used to weight the random or real neighbors i.e. $kwm = \{0.05, 0.1, 0.25, 0.5, 0.75, 1, 1.5, 2\}$. $n = 0.2N$, where $N = 5000$ which is the typical neighborhood size for LIME. $\alpha$ was found for 10 random examples and the maximum of those values was used across other examples in the dataset. Real test neighborhoods are not natural to find for random perturbation based neighborhoods for CIFAR10 as superpixels may vary between neighbors. Hence, GI and CI for CIFAR10 are only reported for realistic neighborhoods generated by Melime. In this case our procedures are also run on these real neighborhoods. More details about explanation generation are in the supplement. We used 56 core and 242 GB RAM LINUX machines to run the experiments.

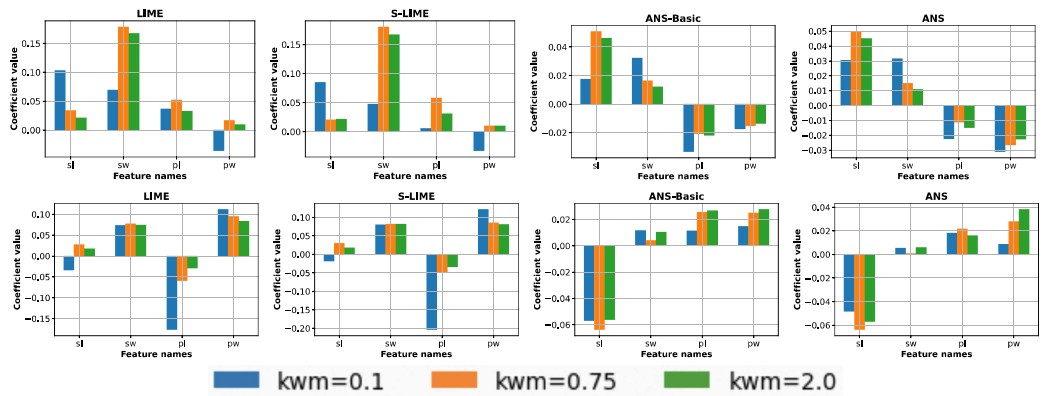

Figure 4: For two examples in the Iris test set corresponding to the two rows of figures above, we provide feature attributions for sepal length (sl), sepal width (sw), petal length and petal width (pw) for three different kernel widths (0.1, 0.75, 2.0). Each row is a separate example and each column corresponds to the method indicated in the title of the subfigure. We can see that across different kernel widths ANS-Basic and ANS feature attributions are much more similar than those seen for LIME or S-LIME. As such, LIME and S-LIME feature attributions even seem to change signs in some cases, while ours do not.

**Observations:**

We see from Figure 2 that ANS and ANS-Basic in general are significantly more accurate than the competitors across kernel widths. ANS while being similar in accuracy and even sample efficiency to ANS-Basic is significantly better in query complexity. The sample efficiency can possibly be improved by considering a larger $n$. The only places where we seem to be worse than the competitors are on GI and CI for the HELOC dataset. We believe the reason for this is that the classification task is much harder for this dataset (state-of-the-art accuracy only 78%) as compared with the others (> 95%) possibly resulting in a highly non-linear decision boundary. This makes the explanations even for neighbors different than the original example, resulting in higher GI and CI values even if the explanation for the particular example is correct. The correctness of the local explanation is reflected by us having a lower INFD value, although GI and CI maybe higher.

The average time taken by LIME for each explanation is about 0.5 seconds, while our methods take about 1 second for the tabular datasets which use random forests. However, for CIFAR10, LIME using ResNet-18 takes about a minute while ANS takes around 50 seconds showcasing the benefit of reduced query complexity. With deeper models this is slated to improve even further. In any case, additional time for our methods in some cases is probably worth it given the higher accuracy and the fact that local explanations are independent and hence can be parallelized.

Qualitatively too in Figures 3 and 4 we observe that ANS produces much more stable and intuitive explanations compared with other methods. ANS-Basic is slightly less stable than ANS but still better than the competitors.

## 5   Discussion

Rather than piecewise linear regression one could also perform piecewise polynomial regression for decision boundaries with high curvature, which will be more computationally expensive, but our rejection sampling method and analysis should be applicable in that context too.

Our proposed approach is distinct from typical rejection sampling schemes [3] where a proposal distribution is *tilted* and used to sample from the desired distribution. The reason we do not subscribe to these standard schemes in our case is that these schemes are mainly used in two settings: i) when we are not able to sample from the original distribution or ii) when we want to sample from the tails of the original distribution. Both of these conditions are not met in our case since we sample in regions which include the mean (i.e. highest probability density) while having access to these distributions.

There are also motivations for our approach based on causality and practical behavior of neural networks.

**Causal motivation:** It is known that given a structural causal model (SCM) [22], the best sparse model captures the Markov blanket – i.e. parents and children in a causal graph – of the variable to be estimated. In the post-hoc explanation setting the target variable $Y$ has no children as the black box model is of the form $y = f(x)$. Hence, if the black box i.e. $f(.)$ is linear lasso it could recover the causal parents for some regularization parameter. However, in the non-linear setting, one could have a case where multiple linear pieces explain the causal relationship in different parts of the domain. If one tries to fit a linear model, gradient for one piece with respect to one feature might (approximately) cancel the gradient with respect to another piece for the same feature because the weights are of opposite signs. This could lead the linear model missing some (causal) parents when explaining the variance in $y$. However, if we are able to identify the correct (linear) regions, then a simple lasso-like fit in each such region should be able to uncover the correct causal parents overall. This work may also motivate a notion of *locally causal*, which may be useful in practice, beyond the standard formalisms of causality [22, 27] which are predominantly global.

**Simplicity bias motivation:** In a recent paper [28], investigating the reasons for neural networks fitting to spurious correlations in the in-domain data resulting in poor generalization, the authors argue that this is because of the simplicity bias of neural networks. That is, the networks pick the simplest boundary to separate classes which could simply be a linear separator on one feature. However, in the test data, the optimal separator could be based on a more complex decision boundary. Hence, one would ideally want to capture the true complexity of the decision boundary. The definition of complexity they use to analyze arbitrarily complex decision boundaries is closely related to the number of linear pieces that would make up different decision boundaries. This formalization thus further motivates our ANS and ANS-Basic approaches, which identify the appropriate linear component based on a piecewise linear decision boundary.

The explanations that we obtain are more accurate in many measures with lower sample and query complexity than standard baselines as we discuss in the experiments. However, like other black box posthoc explainability methods it is difficult to say with certainty that the provided explanations are in fact the "true" explanations. Additionally, the performance gains of our approach are contingent on the quality of the MPLSR schemes. In case the ranges are incorrect the estimation may reduce to standard LIME quality explanations. Luckily, this is likely to only happen when the ranges are overestimated and we include neighbors belonging to different linear pieces, since in the opposite case of underestimation we should still have neighbors belonging to the correct linear piece. In general though, the MPLSR implementations seem to be reasonably mature which is reassuring.

In summary, we have proposed a novel neighborhood generation approach that is adaptive and is robust across different kernel widths, much more so than other methods. We have theoretically shown the benefit of this scheme compared with naive implementation of our idea where our main proposal is likely to be sample but more so query efficient. This is further validated through real data experiments where our approach is more accurate quantitatively as well as qualitatively.

## Acknowledgement

We would like to thank Sara Magliacane and Kartik Ahuja for helpful discussions at the beginning of this project.

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
