## A  Additional Experimental Details

### A.1  Dataset and Neighborhood Generation Details

**IRIS:**  This classic dataset has 150 instances with four numerical features representing the sepal and petal width and length in centimeters, and was obtained from `https://archive.ics.uci.edu/ml/datasets/iris`. The three classes (species of Iris flowers) are: *setosa*, *versicolor*, and *virginica*. A random forest classifier was trained with a train/test split of 0.8/0.2 and yielded a test accuracy of 100%. We provide local explanations for the prediction probabilities for class *setosa*. Experiments were done on the test set with neighborhood size $N$ set to 5000.

**HELOC:** This dataset is obtained from `https://community.fico.com/s/explainable-machine-learning-challenge?tabset-3158a=2`. The HELOC dataset had about 10000 instances, with some records excluded due to empty features. The train/test split was 0.8/0.2 and yielded a test accuracy of 78% using a random forest classifier. We provide local explanations for the prediction probabilities for class 1. Experiments were done on the test set with neighborhood size $N$ set to 5000.

**CIFAR10:**  This dataset has $32 \times 32$ colored images belonging to 10 different classes. The dataset has 50,000 train and 10,000 test samples. The task is to classify these into 10 classes corresponding to dog, bird, and so on. A residual network with 18 units (ResNet18) was trained with test accuracy of $\sim$ 95%. Explanations are generated for the prediction probabilities corresponding to the predicted class for each example. We choose 1000 test examples to generate explanations. Realistic perturbations were generated using VAEGen [2], a Variational Auto Encoder (VAE) fitted on the training dataset. The neighborhood size was 500.

## B  Additional Experiments for Different $n$ Values

Experiments with $n = 0.1N, \ 0.2N, \ 0.5N$ are shown in Figures 5, 6 and 7 for the IRIS, HELOC and CIFAR10 datasets respectively. One common theme seems to be that the INFD is consistently better for both ANS schemes over the competitors. GI is also significantly better on IRIS and CIFAR10. For HELOC it is slightly worse possibly indicating that the neighboring examples found for it may not belong to the same linear piece. This is possible since the nearest neighbors are found just based on the input features. CI seems to be similar for all methods. On CIFAR10 it seems to be much better for the ANS schemes indicating the stability of our approach for image data. The ASC seems to increase with $n$ for ANS over ANS-Basic on CIFAR10 as we find the optimal $\alpha$ changing from 1 to 0.75 to 0.5 as $n$ increases from $0.1N$ to $0.2N$ to $0.5N$ respectively. The smaller $\alpha$ leads to sampling more in the estimated range leading to more samples being accepted and hence higher ASC. QC on the other hand, as expected, is significantly better for ANS over all other baselines and ANS-Basic. This clearly showcases the benefit of our approach in terms of having much higher query efficiency, while still being faithful and stable across kernel widths (a.k.a. different $\sigma$).

## C  Additional Example Explanations

More qualitative examples for CIFAR10 are shown in Figure 8. These further confirm the accuracy of our approach for different kernel widths compared with other approaches.

## D  Societal Impacts

In general local explanations of black box models are beneficial to society where individuals are subject to algorithmic decisions, since they provide transparency and allow for some level of recourse. There are also negative impacts however. Explanations of black box models can be used by adversarial actors to understand the internals of the system and hence game the machine learning system. This problem is exacerbated when accurate explanations can be obtained with low query and sample complexity like we propose in this paper. Other risks include loss of intellectual property for owners of black box models, and privacy concerns. All these can be mitigated by monitoring explanation

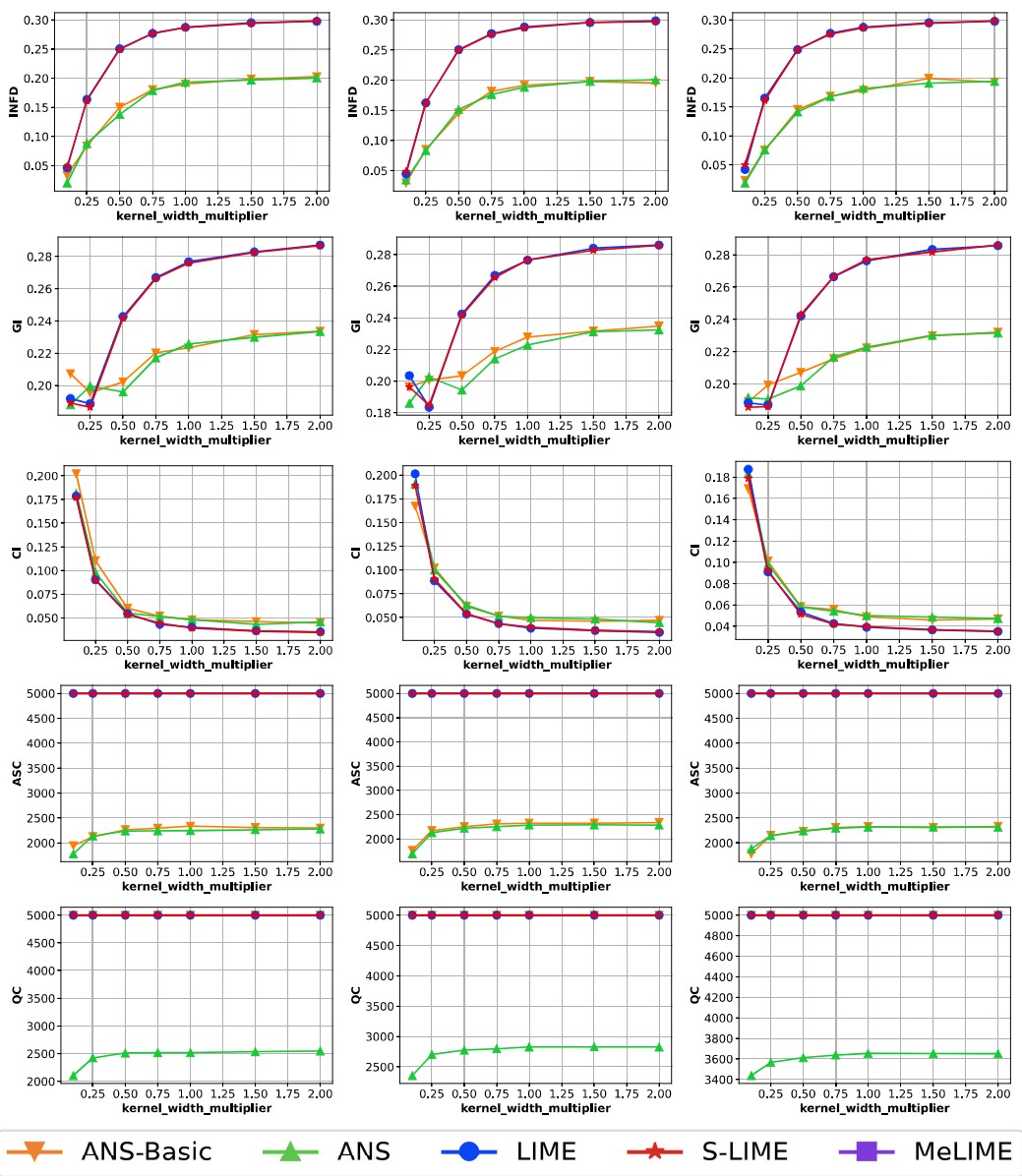

Figure 5: Various metrics vs. kernel width multiplier (kwm) for Iris with $n = 0.1N$ (left), $n = 0.2N$ (middle), and $n = 0.5N$ (right). The metrics are (from top to bottom - each row): Infidelity (INFD), Generalized Infidelity (GI), Coefficient Inconsistency (CI), Accepted Sample Complexity (ASC), and Query Complexity (QC). The method legend is given in the bottom.

requests and putting in place some safeguards around throughput for explanations. Many of these risks are common to other local explainability methods as well and call for research in this area.

## E   Figures with Error Bars

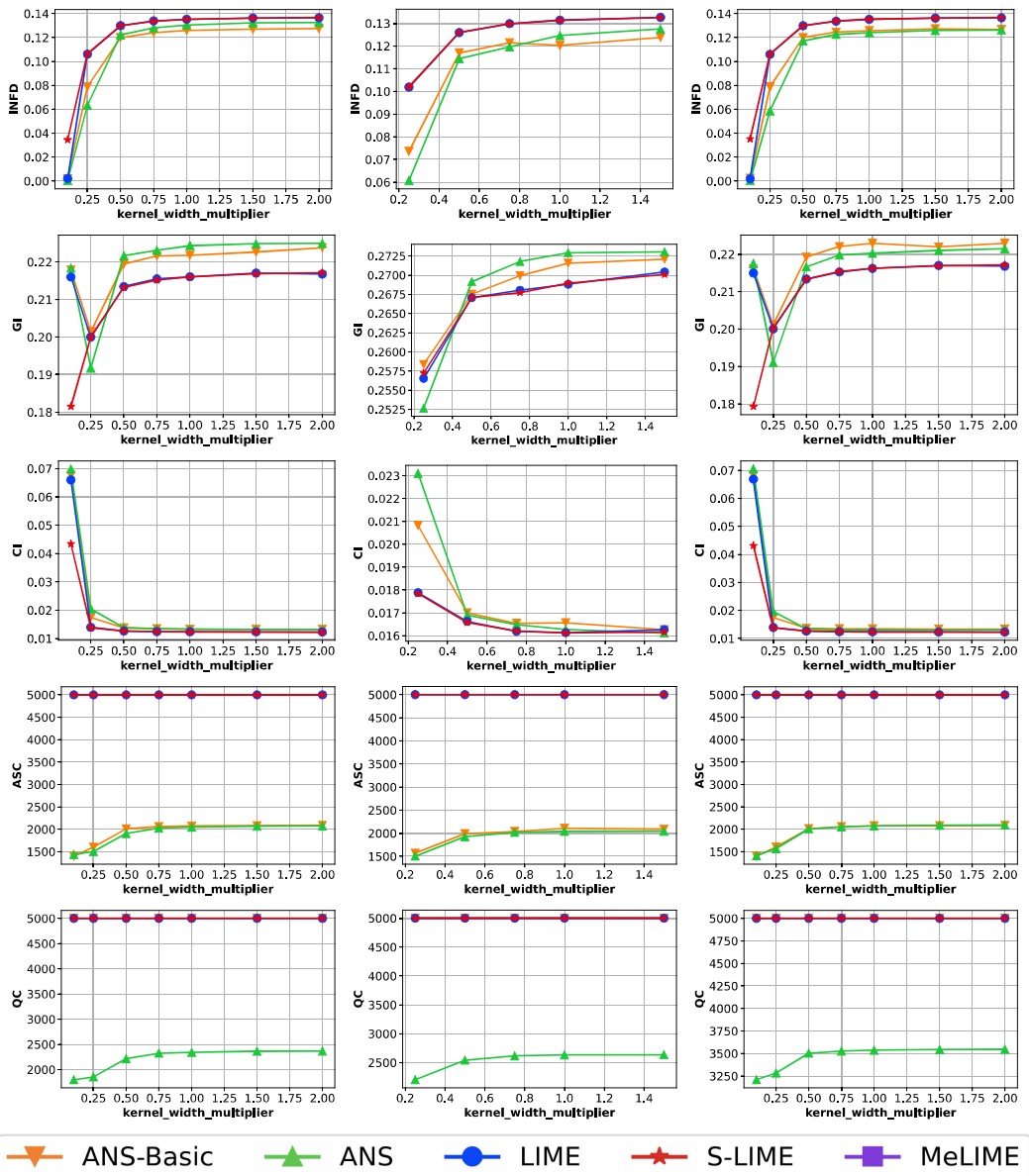

Figure 6: Various metrics vs. kernel width multiplier (kwm) for HELOC with $n = 0.1N$ (left), $n = 0.2N$ (middle), and $n = 0.5N$ (right). The metrics are (from top to bottom - each row): Infidelity (INFD), Generalized Infidelity (GI), Coefficient Inconsistency (CI), Accepted Sample Complexity (ASC), and Query Complexity (QC). The method legend is given in the bottom.

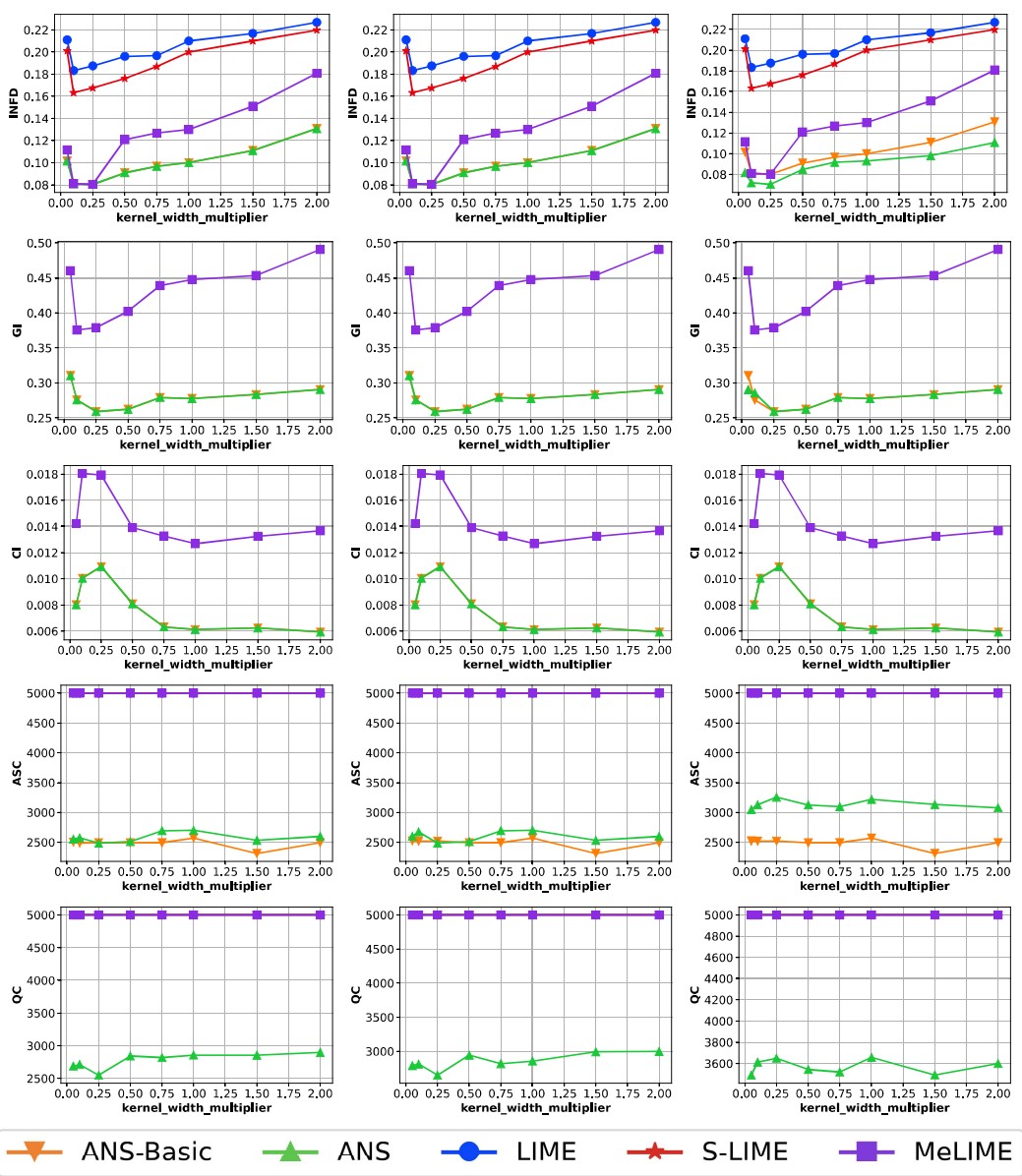

Figure 7: Various metrics vs. kernel width multiplier (kwm) for CIFAR10 with $n = 0.1N$ (left), $n = 0.2N$ (middle), and $n = 0.5N$ (right). The metrics are (from top to bottom - each row): Infidelity (INFD), Generalized Infidelity (GI), Coefficient Inconsistency (CI), Accepted Sample Complexity (ASC), and Query Complexity (QC). The method legend is given in the bottom.

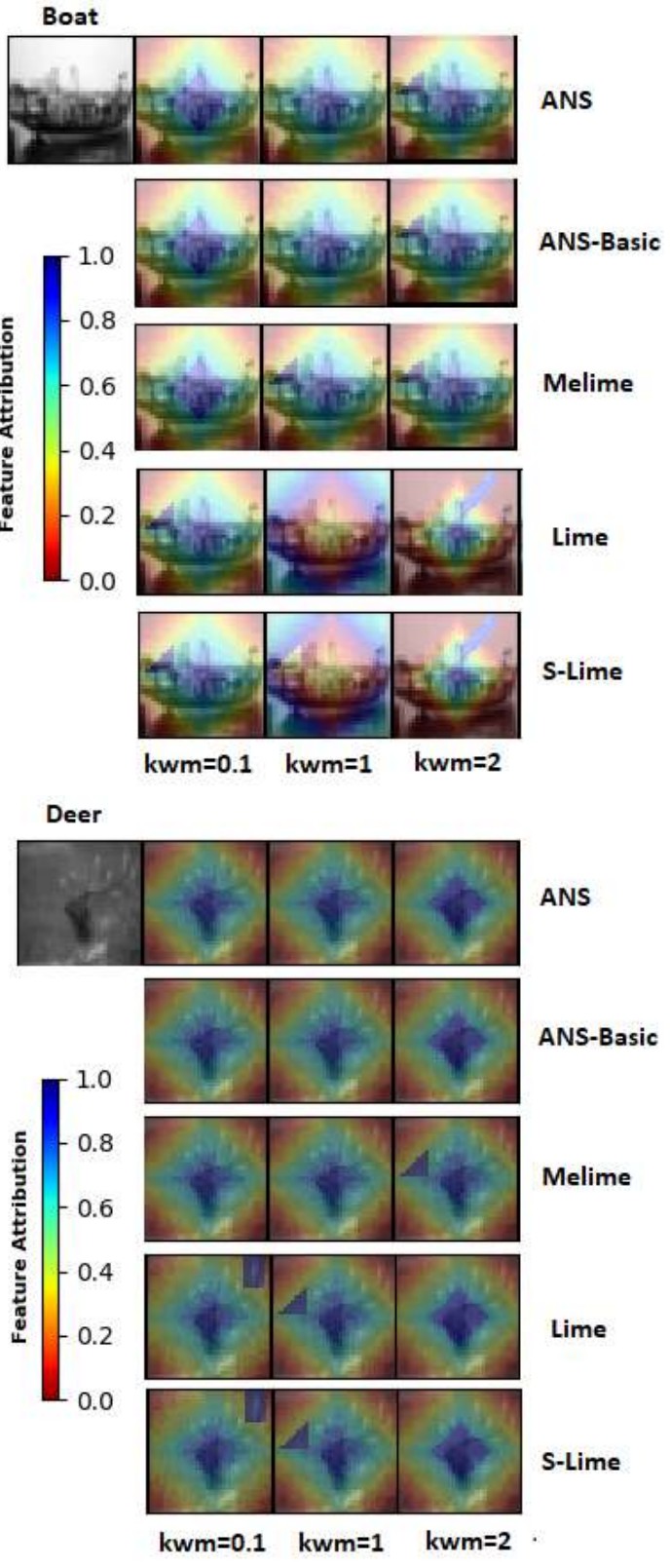

Figure 8: More qualitative explanations for images from the CIFAR10 dataset for three different kernel width multiplier (kwm) values. The feature attributions of our proposed method ANS (and even ANS-Basic) are much more stable and accurate over different kwm values compared to other methods as also seen in the main paper.

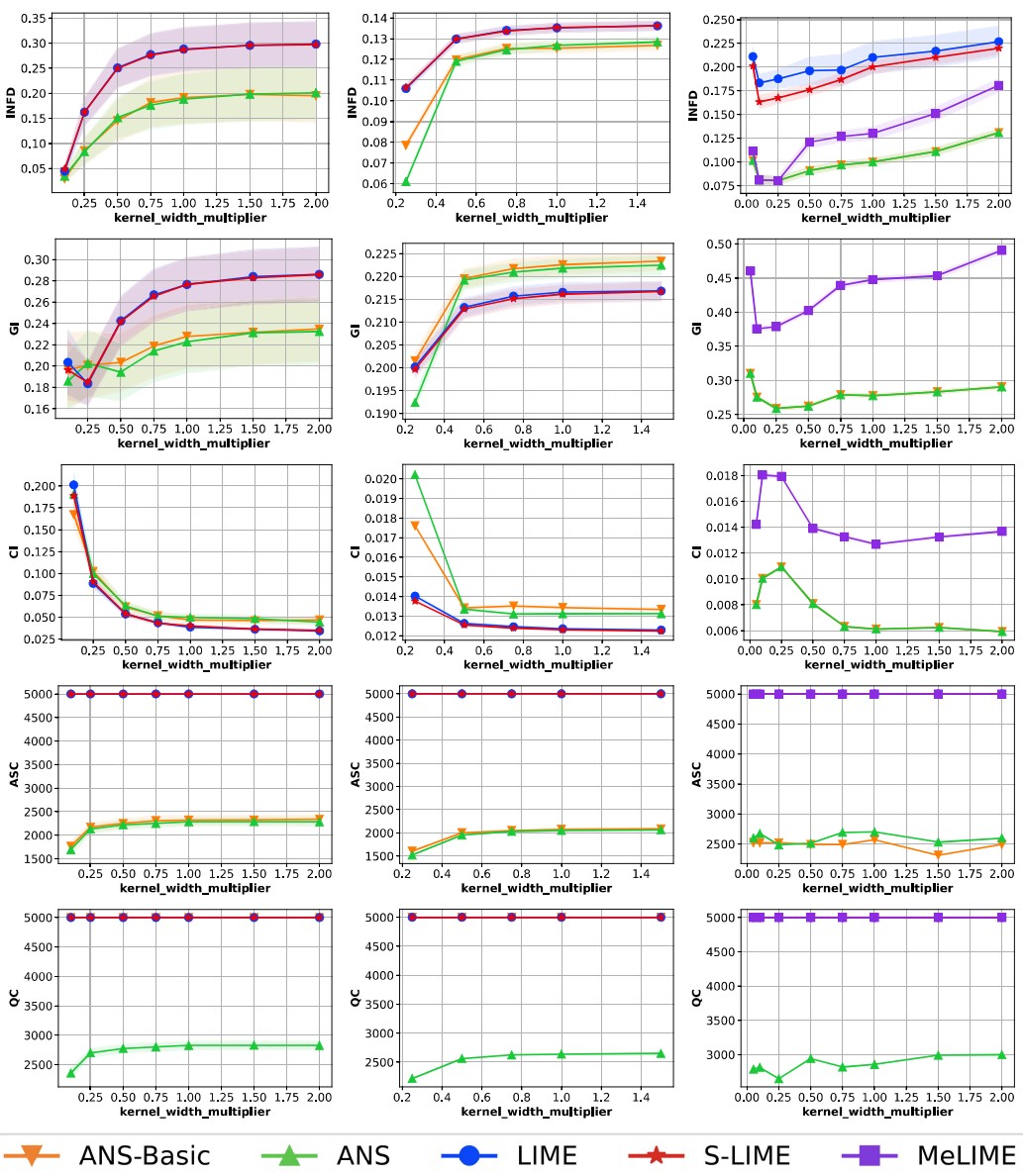

Figure 9: Various metrics vs. kernel width multiplier (kwm) for Iris (left), HELOC (middle), and CIFAR (right) datasets with $\pm 1$ standard error of mean shading. The metrics are (from top to bottom - each row): Infidelity (INFD), Generalized Infidelity (GI), Coefficient Inconsistency (CI), Accepted Sample Complexity (ASC), and Query Complexity (QC). For all metrics lower values are better. The method legend is at the bottom.