# OpenReview forum: "Is this the Right Neighborhood? Accurate and Query Efficient Model Agnostic Explanations"
_NeurIPS.cc/2022/Conference — NeurIPS 2022 Accept_

### Official Review · Reviewer_15VB · 2022-06-27

**Rating:** 6
**Confidence:** 3
**Soundness:** 3 good
**Presentation:** 3 good
**Contribution:** 3 good

**Summary:**

This paper proposes a technique called adaptive neighborhood sampling scheme (ANS) to make local explanations more faithful, stable, and sample query efficient. ANS is built on multidimensional piecewise linear segmented regression (MPLSR) to indentify boundaries of each linear piece.

**Questions:**

(1) for Cifar10, how do you sample neighbors? in input image space or feature space?

**Ethics Review Area:**

["I don’t know"]

**Limitations:**

(1) It would be better if the author can provide some qualitative examples on the Tabular datasets.

**Strengths And Weaknesses:**

Pros:

(1) The motivation is clear and the method to use MPLSR is solid to identify linear pieces.

(2) Experimental results show the better fidelity and sample efficiency of the proposed methods.

Cons:

(1) I think this paper needs to give a formal definition of "stability" and "faithfulness".

(2) It would be easier for readers to understand the metrics if the author can explicitly indicate for each metric whether it is higher the better or lower the better.

(3) According to figure 2, for HELOC dataset, it seems that ANS-Basic and ANS are worse than the baselines wrt GI and CI. Also for CIFAR-10, a comparison to LIME is missing.

---

> ### Author Response · Authors · 2022-07-30
> **Thank you for your comments...**
>
> **Definition of stability and faithfulness:**
>
> -- We now have defined both of these terms at the end of the introduction.
>
>
> **indicate for each metric whether higher or lower is better:**
>
>  - We now have indicated in Figure 2 caption that lower values for all metrics are better.
>
>
> **for HELOC dataset, it seems that ANS-Basic and ANS are worse than the baselines wrt GI and CI:**
>
> -- We now have discussed this point in the Observations part of Section 4.
>
>
> **for CIFAR10 comparison with LIME is missing:**
>
> -- LIME is present in blue for INFD. It is also present for ASC and QC, but just overplotted by MeLIME since both use all samples and hence have same complexities. We could not compute GI and CI for it since LIME uses superpixels for images which are different for each image and hence it is not possible to apply the same model to neighbors and obtain GI or compare coefficients since the features (i.e. superpixels) themselves are different.
>
>
> **for Cifar10, how do you sample neighbors? in input image space or feature space?**
>
> -- For ANS we sample in the input space where each pixel is a feature.
>
>
> **Tabular data qualitative examples:**
>
> -- Figure 8 in the supplement provides some (randomly selected) qualitative examples for the IRIS dataset, where we again witness the stability of our method.

---

> > ### Comment · Reviewer_15VB · 2022-08-07
> > **Most of my concerns are addressed**
> >
> > Most of my concerns are addressed. Are other works for interpretability also sample neighbors in the input pixel space for cifar-10?

---

> > > ### Author Response · Authors · 2022-08-07
> > > **Thank you for responding**
> > >
> > > We are glad that most of your concerns have been addressed. Yes, most other methods sample in input space. Even manifold methods such as MeLime which sample in the latent space end up decoding these neighbors and finally fitting a (proxy) explanation model in the input space, where the explanations highlight important input pixels since the explanations have to be humanly understandable.

---

### Official Review · Reviewer_6LhD · 2022-07-11

**Rating:** 6
**Confidence:** 2
**Soundness:** 3 good
**Presentation:** 3 good
**Contribution:** 3 good

**Summary:**

The paper proposes a rejection sampling method for neighborhood-based explanation methods including LIME. It argues that existing neighborhood-based explanation models are quite unstable in terms of the the neighborhood width, which is the boundary range for the local neighborhood and the mean value of a Gaussian distribution where the local neighborhood follows. In order to make the choice of neighborhood width insensitive, it first samples several data points (n) to compute the range $[a_n, b_n]$ and an uncertainty score $\alpha$ to adaptively learn the neighborhood distribution from $N(\mu, \sigma I)$ to $N(\alpha \mu + (1-\alpha)\frac{a_n + b_n}{2}, \sigma I)$. Then the LIME algorithm is applied for fitting an interpretable explainable model. It also analyses the sample efficiency and query efficiency compared to a naive version, showing that the method is query/sample-efficient.

**Questions:**

- It would be nice if the variance can be provided in Figure 2 to show the performance of the proposed method is stably outperforming other methods.


**Limitations:**

The limitation is not discussed in the paper.

**Strengths And Weaknesses:**

- Strengths
    - The proposed method is intuitive and relatively clearly explained.
    - The experiment shows that the method is simple and effective.
    - Efficiency analysis is covered in terms of sampling, query and training MPLSR.
- Weakness
    - The experiments are limited to small-scale datasets.

---

> ### Author Response · Authors · 2022-07-30
> **Thank you for your comments...**
>
> **Report Variances:**
>
> -- We now report standard errors in Figure 9 in the supplement. If you think this is appropriate we will add it to the main paper in the final version. Error bars are larger for IRIS since it is a significantly smaller dataset than the other two (test size = 30).
>
>
> **Limitations not discussed:**
>
> -- Sections F and G in the supplement discuss limitations and societal impact of our approach respectively.
>
>
> **Small scale datasets:**
>
> -- The datasets we used have been commonly used in other explainability works (Ramamurthy et. al. NeurIPS 2020; Dash et. al. NeurIPS 2018; Dhurandhar et. al. arxiv 2022; Arya et. al. JMLR 2020).

---

### Official Review · Reviewer_8rcz · 2022-07-12

**Rating:** 6
**Confidence:** 3
**Soundness:** 3 good
**Presentation:** 3 good
**Contribution:** 3 good

**Summary:**

Many local explanation methods have been developed for explaining black box models in a post hoc fashion. These methods first sample around the example to be explained and build a linear model using the samples to form an explanation. However, the challenge in these schemes is that we do not know if we are sampling the right neighborhood around the example. If the neighbourhood is too small, the linear fitting can be unstable due to a bad condition number and if the neighborhood is too large, the function that these methods approximate may not be linear. This work provides an adaptive sampling procedure where they first estimate the linear region around the example using a few samples and then sample within that region only taking into account the uncertainty of the estimate. This work then shows using experiments on a few tabular datasets, that their procedure is much more query efficient and sample efficient and leads to stable predictions across multiple widths of sampling.

**Questions:**

The related work section does not mention any previous work which also does adaptive sampling of the neighborhood, I wanted to clarify if this is the first work which does the adaptive sampling of the neighborhood.

The stability with respect to kernel widths is shown in the figures in the appendix. I see that Melime is also quite stable across different widths. Can the authors please comment a little more on the Melime method?

I also want to clarify if the metrics shown in the graphs are averaged over all test examples.

**Limitations:**

The authors have adequately discussed the limitations of their work. They have showed that their explanations are much more stable across different sampling widths but as pointed out by the them, there is no way to check if the explanations are more accurate. Moreover, the correctness of their method depends on MPLSR schemes and if the range of the linear segment is found to be incorrect in these schemes, the quality of explanations may degrade. They have also pointed out that their methods is slower as compared to the vanilla methods and have also given a dataset example where their performance is weaker than the vanilla methods.

**Strengths And Weaknesses:**

The method proposed in this work seems like a very natural and simple idea which gives improvement over the existing methods. The explanations are stable over different widths which is what was desired.

One weakness is that this method is slower because the method involves running MPSLR schemes and the accuracy of the method depends a lot on the correctness of MPSLR methods. The method is also somewhat complicated to implement and involves choosing multiple hyper parameters like n, \alpha.

The paper is generally well written and clear.

There could be more discussion on the related work. It would be nice if there were more details on how different methods achieve stability in their methods. Also, stability and faithfulness have been mentioned multiple times but it is not clearly defined anywhere what do the authors mean by that. I am guessing stability is defined as the change in explanations as the sampling width increases. But, in any case, these terms should be clearly explained in the beginning.

---

> ### Author Response · Authors · 2022-07-30
> **Thank you for noticing that ANS might be the first adaptive sampling scheme for local explanations...**
>
>  **method is slower:**
>
> -- This actually is not necessarily the case. This was in the supplement but now we have mentioned in the observations subsection (in Section 4) in the main paper that for ResNet-18 we are actually faster than LIME. The reason for this is that as NN models get deeper inference time is no longer insignificant and hence having to query it fewer times more than compensates for the extra running time of MPSLR schemes. Even for the ResNet-18 case we were about 10 seconds faster per example. This gap should increase as we consider deeper models.
>
>
> **involves choosing multiple hyper parameters like $n$, $\alpha$:**
>
> -- You are right that $n$ may have to be chosen, however $\alpha$ can be estimated per equation 1. Even $n$ could be chosen by seeing performance for different values and seeing the behavior as shown in Figures 4-6 in the supplement.
>
>
> **accuracy depends on correctness of MPSLR methods:**
>
> -- As discussed in section F of the supplement, although we want reasonable MPSLR schemes we are mainly affected by error only in one direction. As such, only overestimation of the range (underestimation of the number of linear pieces) is a problem as it may cover non-linearities however, if we are to underestimate the range (overestimate the number of linear pieces) it is largely fine since the black box will still correspond to the correct linear piece in this case.
>
>
> **Defining stability and faithfulness:**
>
> -- We now have defined both of these terms at the end of the introduction.
>
>
> **I wanted to clarify if this is the first work which does the adaptive sampling of the neighborhood:**
>
> -- Thanks for noticing this, which even skipped our minds. Yes, you are right to claim that this might be the first adaptive neighborhood sampling procedure for local black box explainability.
>
>
> **Stability of MeLIME:**
>
>  -- Although we are still more stable than MeLIME it is seemingly stabler than other alternatives in some cases. We believe this happens because MeLIME uses realistic perturbations which leads to a better neighborhood thus resulting in superior explanations.
>
>
> **More details on other methods claiming stability:**
>
> -- We now have added some details for methods that target stability in the related work section as indicated by you.
>
> **clarify if the metrics shown in the graphs are averaged over all test examples:**
>
> -- Yes, the results are over all test samples.

---

> > ### Author Response · Authors · 2022-08-08
> > **Any further clarifications?**
> >
> > Since the response period ends tomorrow. Please let us know if you have any further questions/concerns. Thank you.

---

### Official Review · Reviewer_hHAF · 2022-07-17

**Rating:** 4
**Confidence:** 2
**Soundness:** 3 good
**Presentation:** 3 good
**Contribution:** 2 fair

**Summary:**

This paper is on improving explainability techniques for machine learning models. A popular technique is based on local linear models that make a locally linear approximation of the ML model around a provided input example for which an explanation is sought. Local linear approximations make use of random samples from around the provided input example to estimate the linear model. However, having samples too close to the input can affect the condition number of the covariance matrix, while on the other hand having the samples spread out too much can lead to an unfaithful model.

This paper proposes a strategy that chooses the samples in an adaptive manner to ensure that the samples span the locally linear region of the model, leading to a more reliable model. Experimental evaluation shows better performance compared to other locally related baselines.

**Questions:**

NA

**Strengths And Weaknesses:**

**Strengths**:

The proposed method samples points adaptively thus not requiring to set a sampling variance.
The method also takes into account the uncertainty of the estimated values a_n, b_n for improved robustness. Experimental evaluation shows that it leads to a better performance w.r.t. to a variety of performance metrics.

**Weaknesses**:
I do not work in explainability for ML, however it seems that the contribution is a minor improvement (the sampling strategy) over existing approaches. As such, the approach is not so novel, in my opinion, even though experimentally it is beneficial.

---

> ### Author Response · Authors · 2022-07-30
> **We believe our contribution to be conceptually and methodologically significant...**
>
> **minor improvement over existing approaches:**
>
> -- As correctly pointed out by reviewer 8rcz ours is to the best of our knowledge the first adaptive sampling method for local post-hoc black box explainability which is one of the hottest topics in XAI research as it can have significant bearing in appropriating trust in models (Ferrario et. al. FAccT, 2022; Arya et. al. JMLR 2020). As mentioned in the introduction, sampling is the most critical part for generating faithful and stable explanations, since the explanation model itself is fixed (e.g. sparse linear). All other works (mentioned in the paper) have static sampling strategies that are either in the input space or on a manifold. Conceptually, we are the first to realize that manifold or not examples sampled in a neighborhood could belong to different linear parts in a non-linear function such as a deep ReLU network and hence using them to obtain a local explanation through sparse linear or some other simple model fitting can be misleading (mentioned in related work). Theoretically and operationally, we show that our adaptive sampling method can be significantly more query (and sample) efficient which has not just computational but direct monetary significance in today's multi-cloud world as pointed to in the introduction. Moreover, we believe the simplicity of the solution to be a positive as it can be easily implemented and hence more widely used. Equally importantly, this solution was arrived at after giving significant thought to other alternatives such as rejection sampling where one typically has a proposal distribution belonging to the natural exponential family (NEF) that one uses to sample from. However, this option was thoughtfully rejected by us given the two factors mentioned in the Section 5. Hence, although our proposal may seem simple it requires thought, analysis, and careful experimentation to arrive at. Not to mention our approach besides being intuitive has strong motivations grounded in causality and practical behavior of neural networks, which we have now added in Section 5. In the original submission, these were mentioned in the supplement. We hope that these points convince you of our contribution. Thank you.

---

> > ### Author Response · Authors · 2022-08-05
> > **Looking forward for your response...**
> >
> > As argued in our response above we believe our contribution to be significant which the other 3 reviewers also seem to be in agreement with. Please let us know if you have any other questions. Thank you.

---

### Author Response · Authors · 2022-07-30
**Common response**

We thank all the reviewers for their constructive comments. We are glad that you found our paper to be well-written, intuitive and novel. Reviewer 8rcz even pointed out (which we missed) that our work might be the first adaptive sampling work in this area. Thanks for recognizing this.


Based on the reviewer comments we have made the following updates to the paper. Note that these updates are highlighted in *blue* in the updated paper.


Main paper updates:
1) We have now clarified what we mean by faithfulness and stability.
2) We have added more details regarding related works that discuss stability.
3) We have indicated that lower is better for all the metrics in the experiments.
4) More discussion of observations added in the experimental section.
5) Moved section B from the supplement to section 5 in the main paper which provides causal as well as a neural network behavioral motivation for our approach.


These were the major updates to the paper. We now address individual reviewer comments.

---

### Meta-Review · Area_Chair_ndAr · 2022-08-26

**Recommendation:** Accept
**Confidence:** Less certain

**Metareview:**

The paper attacks the problem of how to define "local" when generating local linear explanations (e.g. LIME).  Forming the linear approximation using multiple points, the proposed method attempts to balance robustness of the explanation vs its specificity.  The approach of using multidimensional piecewise linear segmented regression is sensible for this end, albeit if increasing runtime.  The majority of reviewers had a favorable opinion of the work, recognizing the paper's contribution as targeted but important, given the popularity of local linear explanations.  Even reviewer hHAF, who recommended rejection, recognized the work's practical benefit ("experimentally it is beneficial").  Thus, I recommend acceptance.

**Award:**

No

---

### Decision · Program_Chairs · 2022-09-14

Accept